# Improved dielectric performance of graphene oxide reinforced plasticized starch

**Eashika Mahmud, Shafiqul I. Mollik, Muhammad Rakibul Islam** [ID] *

Department of Physics, Bangladesh University of Engineering and Technology (BUET), Dhaka, Bangladesh

* rakibul@phy.buet.ac.bd

## Abstract

High dielectric constants with less dielectric loss composites is highly demandable for technological advancements across various fields, including energy storage, sensing, and telecommunications. Their significance lies in their ability to enhance the performance and efficiency of a wide range of devices and systems. In this work, the dielectric performance of graphene oxide (GO) reinforced plasticized starch (PS) nanocomposites (PS/GO) for different concentrations of GO nanofiller was studied. The surface morphology, and chemical and structural properties of the PS/GO nanocomposites were investigated by field emission scanning electron microscopy (FESEM), Fourier-transform infrared spectrometry (FTIR), and X-ray diffractometer (XRD). The FESEM study showed a uniform dispersion of the GO nanofiller in the nanocomposites. The XRD analysis showed a reduction in d-space due to the incorporation of GO nanofiller in the nanocomposites. The FTIR data exhibits the formation of hydrogen bonds among PS and GO nanofillers, suggesting the presence of strong interaction between them. The dielectric properties of the nanocomposites were studied at room temperature in the frequency range 100 Hz–1 MHz. The dielectric constant was found to improve due to the incorporation of GO. This composite nanomaterial also provides low dielectric loss at low frequency. Moreover, an increasing trend is observed for the AC conductivity of the composites. From the complex impedance study, the changes in various impedances with low to high-frequency ranges have been calculated and explained in the equivalent circuit diagram. The complex impedance spectra analysis shows the change in resistance and constant phase element (CPE): grain boundary resistance, $R_2$ decreases from 4.3 KΩ to 1.9 KΩ, and CPE increases from 0.59 μF to 0.72 μF for PS/GO (0.5%) nanocomposite. This study will provide a potential route for the fabrication of biocompatible dielectric device fabrication.

## 1. Introduction

The electronics and energy storage industries are progressing where electronic circuits have to adapt by becoming smaller, more integrated, lightweight and smarter to keep pace with globalization trends alongside green electronics [1]. Here biopolymers as dielectric materials are leading as increased interest not only because they offer advantages such as low dielectric loss

**Data Availability Statement:** All the data files are available from the figshare database DOI: 10.6084/m9.figshare.25952173

**Funding:** The author(s) received no specific funding for this work.

**Competing interests:** he authors have declared that no competing interests exist.

and high breakdown strength but also it has practical advantages over heavy metal, conventional silicon di-oxide device, ceramics or synthesized polymer along with easy processing and diverse structures, low weight [2, 3]. Among various industrial applications, they have advancements as insulators, particularly used in organic field effect transistor (OFET), wind turbine, aerospace, memory storage, energy storage, wastewater remediation, and so on [4–6]. However, biopolymer must have a high dielectric constant that can withstand capability with high electric field, and thermo-mechanical stability, high dielectric constant and less dielectric loss. Particularly in energy storage devices, as energy is directly proportional to the dielectric constant and the electric field, building energy storage devices with a high dielectric constant can provide more energy to store [7–9]. A single biopolymer has low dielectric constant with poor thermal stability, which has limited its application. These are the reasons why there has been a surge in research aimed at developing polymer-based dielectric composites with superior properties, including high dielectric constant, low dielectric loss, and high breakdown strength simultaneously. These aforementioned advantages are warranted in organic layer polymer [10–12].

Among biopolymers, it is hard to find polymers that are easily synthesizable, water-soluble and degradable under soil, which can conserve the ecosystem. Although, a variety of biopolymers including polysaccharides, gelatin, cellulose, polylactic acid (PLA), chitosan, lipid or hybrid biopolymer, have been used as the matrix [13]. In particular, among the natural polysaccharide, plasticized starch (PS), is considered a prominent one because of the fulfillment of the aforementioned criteria. Starch is mostly derived from plants. Tubers of potato, cassava, and corn are the most available and starch-enrich among them. Potato is cheap, easily found in the local market, easy to extract, and like other sources, it dissolves in both water and soil [14]. These advantages make it a popular choice as a perfect alternative of another polymer. For example, carboxymethyl cellulose (CMC) shows low resistivity to insects and fungi and is sensitive to light, even easily distorted by enzymatic corrosion. Then, chitosan is soluble in an acidic solution, thus pH controlling is an issue in its synthesis [15]. Starch hasn't the aforementioned drawbacks. However, like the common drawbacks of biopolymers, starch is poorly thermo-mechanically stable with low dielectric constant. This shortcoming can be overcome by modification with clays, nanoparticles, crosslinking, organic-inorganic reinforcement, or cellulose fiber [16].

The material optimization process with incorporation of fillers into the matrix is becoming a more practical and efficient way of heading to the better and revolutionizing the target application. Since 2004, the first isolation of a single layer of graphene has gained vast attention due to its high mechanical and thermal stability [17, 18]. However, a perfect single monolayer synthesis is naturally difficult. Hence, graphene oxide (GO) is an intermediate two-dimensional (2D) chemical product of graphene. Unlike graphene, it contains oxygen-containing functional groups: carboxyl, hydroxyl, epoxy, carbonyl etc. [19]. Van der Waals' interaction with polymer eases the solution process and makes the synthesis process easy, simple, and cost-effective. These functional groups on the surface with the chemical bonding capacity of carbon make it easy to create hydrogen bonds, easily dissolve in water, and ultimately make them hydrophilic. This behavior makes GO compatible with biopolymers, and GO easily swells. Thus the synthesis becomes easier [20]. So GO is uniformly dispersed in the PS matrix even at a comparatively low temperature with a high aspect ratio. Thus, PS/GO creates a unique layer structure, improves the dielectric property and the shortcomings of mechanical properties, and provides thermal stability [21].

Many research groups have been working on graphene oxide-reinforced starch. Rapisarda et al. prepared starch/GO gel hybrid binder and observed high specific capacitance, which is applicable in supercapacitors as bio-binders [22]. Carlos et al. prepared corn starch-GO

nanocomposite through the melt mixing method, where their highly dispersed irregular structure is observed with improved thermal-mechanical stability [23]. Li et al. also observed the enhancement of the mechanical and thermal performance [24]. Zheng et al. prepared graphene-starch nanosheets (GN-S), then GN-S/PS nanocomposite, by which UV absorbance was improved, and according to their suggestion, the possible application is UV protector [25]. Wo et al. prepared plasticized starch/graphene oxide (PS/GO) with polylactic acid (PLA) and proposed PS/GO as a compatibilizer for starch/PLA composite [26]. Bhattacharyya et al. also prepared a cross-linked PS/GO composite focused on the decontamination of water [27]. Reported work on starch-based materials so far: improvement of mechanical, electrochemical, optical, and thermal properties [28–30]. To the best of our knowledge, the dielectric performance of PS/GO is still being studied. Therefore, this work is fully intended to investigate the complete dielectric properties where the influence of GO on the PS matrix is keenly focused.

Based on economic demand, easy processing, and cost-effectiveness are one of the main focuses of this work. In this work, we have used the solution casting technique. To check the dispersibility and the effect of molecular structure, we've done FTIR, XRD and FESEM. The thorough study of dielectric performance: dielectric constant, dielectric loss, AC conductivity, and complex impedance spectroscopy analysis have been studied to understand their behavior with frequency and polarization domain. As per our observation, these PS/GO fabricated nanocomposites are expected to be suitable for the insulator device. In the large view of climate sustainability, this composite is completely biocompatible and ensures low cost- high yield production.

## 2. Materials and characterization

### 2.1. Materials

Graphite powder (flake), potassium permanganate ($KMnO_4$, Merck, India), sodium nitrate ($NaNO_3$), concentrated sulfuric acid ($H_2SO_4$, 98%), hydrogen peroxide ($H_2O_2$, 30%, Qualikems, India), di-ionized (DI) water, glycerol (Propane-1, 2, 3-triol), and vinegar were purchased for this study. All the reagents were of analytic grade. Fresh potato is collected from the local market, and powder starch is extracted in the laboratory.

### 2.2. Synthesis procedures

**2.2.1. Starch extraction from potato.** Potatoes were first collected from the local market. Then, it was cleaned, peeled, and grated carefully. The grated potato was put into a beaker with DI water. Then, the potato mixed water was stirred with a glass rod. Before the liquid color became brown, the grated potato was separated from the liquid. Rest the beaker steady until the thick starch powder settles at the bottom of the beaker. Separated the sediment starch by using a water purifying sediment technique, but in this process, white starch is collected instead of water. With the help of a spatula, the starch cake is shifted to a petri dish and placed in an oven for drying. The wet starch is dried for a few hours at (55–60)˚C. The well-dried powder is converted to fine powder by mortaring it using mortar-pastel.

**2.2.2. Synthesis of GO.** Graphene Oxide (GO) was synthesized from graphite powder by applying modified Hummer's method [31]. For the preparation of graphite solution, a 5g graphite flake mixed with 2.5 g Sodium Nitrate ($NaNO_3$) and 115 ml sulfuric acid in a beaker was stirred with a glass rod and kept in refrigerator. Then, the beaker was kept in a saucepan, where some ice cube was put so that the temperature could be kept below 20˚C. Afterward, the slow addition of 15g $KMNO_4$ and continuous stirring changed the solution color from black to olive green. Thus, we produced bimetallic heptoxide ($Mn_2O_7$) as a byproduct, which enhances the reactivity. Then, on a hot plate, the dense solution was stirred for 2 hours at

 

35˚C. Now, in the oil bath stage, the solution was kept in an oil-poured saucepan and stirred with the addition of 230 ml DI water slowly at around 90˚C. DI water was used to make sure that GO was completely acid-free. Then, we stirred the solution at room temperature and kept stirring until the temperature was below 35˚C. Again, oil bathing and stirring were done with a temperature below 70˚C, and a slow addition of 230 ml DI water and 50 ml hydrogen peroxide ($H_2O_2$) were performed. Slowly, the addition of $H_2O_2$ made the solution dark brown, which was used in order to minimize the residual manganese dioxide and permanganate. Then, the solution was kept at room temperature overnight to cool naturally until the GO was settled as sediment. The wet GO is centrifuged at 8000 rpm for 15 minutes each with ethanol and then with DI water to remove any exfoliated GO and to obtain a homogenous GO suspension. Finally, to obtain the dry GO, it is dried in an oven at 40˚C.

**2.2.3. Preparation of plasticized starch (PS).** At first, 5g starch powder was mixed with DI water to prepare an aqueous solution. As starch is a long chain microstructure molecule, for breaking the carbons long chain vinegar and propan-1, 2, 3-triol (glycerol) were mixed with the solution. Another reason of using glycerol is to plasticize the solution. To ensure homogenous dispersibility, the solution is stirred with glass and simultaneously heated at 90˚C on a hot plate until the mixture is dense enough to form a film. This dense mixture was then shifted to a petri dish, which was kept in an oven at 60˚C for several hours. When the mixture was formed as the petri dish shape, it was can easily be separated from the petri dish [32].

**2.2.4. Preparation of PS/GO nanocomposite.** Similar to the preparation of PS film, an aqueous solution was prepared. However, for homogenous dispersion, GO powder was first sonicated for about an hour to completely disperse in water. As two different concentrations of GO fillers (0.5 wt% and 1.0 wt%) were sonicated in two different beakers. Then 5 g PS powder, 3 ml glycerol, and 3 ml vinegar were poured into each beaker and mixed with the help of glass rod. The mixture-to-film route follows the same heating process but at 95˚C. Then the PS/GO blends were decanted onto two different petri-dish accordingly and placed at oven for heating at 50˚C until the slurs were formed into film. PS/GO (0.5%) and PS/GO (1%), two different nanocomposite films were made eventually, three in total [33].

## 2.3. Characterization methods

Four different characteristics were used: XRD, FTIR, FESEM, and dielectric properties shortly. The first three are for understanding the composite's combined chemical interaction: microstructure and surface study. The later properties include dielectric constant, dielectric loss, AC conductivity, and impedance spectroscopy, which is the study of individual frequency behavior of the nanocomposites.

X-ray diffractometer, 3040XPert PRO, Philips was used where the material was irradiated with CuKα monochromatic light source. The diffraction pattern was scanned from 5º to 90º at 2 º/min speed. Fourier-transform infrared spectrometry (FTIR) spectrometer: IRSpirit, Shimadzu was used to find the chemical bonding of all 3 films. In the recording, wave numbers range from 4000 cm$^{-1}$ to 400 cm$^{-1}$. Field Emission Scanning Electron Microscope (FESEM): JEOL-JSM 7600 was used to study the fractured, cross-section surface of all 3 films at an accelerating voltage of 5 kV. A portable impedance network analyzer: FieldFox N9923A, Agilent Technologies Inc., was used to study the dielectric properties. The dielectric performance was run at room temperature within $10^2$–$10^8$ Hz frequency range.

 

## 3. Results and discussion

### 3.1. Structural analysis

XRD is mostly the study of diffraction patterns of crystalline materials but is also powerful for organic compound's microstructure. Fig 1 represents the XRD pattern of all three nanocomposites. All three curves have almost identical peaks and denote the nanocomposites are amorphous [34, 35]. The diffraction peaks at 17°, 20°, 22°, and 24° confirm the starch of this work is B type and extracted from potato [36, 37]. For PS/GO nanocomposites, a slight peak intensity decrement is observed at a 22° peak, which may be attributed to the fact of GO incorporation and its rational impact on PS crystalline structure. At around 48°, a slight bump of GO characteristic peak appears, which suggest, a well dispersion. Then, the bump becomes more visible with the increasing concentration of GO, from 0.5% to 1%. Additionally, it's noted that the introduction of GO (0.5%) causes a displacement of peaks towards higher scattering angles, suggesting a decrease in the d-spacing. Conversely, the inclusion of 1.0% GO results in a shift of peaks towards lower angles, indicating an expansion in the d-spacing.

### 3.2. Chemical bond analysis

Fig 2 shows the distinctive chemical bonding and the changes of bonding after the addition-increment of filler concentration with the corresponding resonant frequency. For PS film, the sharp peaks observed at 3352 cm$^{-1}$ and 1651 cm$^{-1}$ correspond to the stretching of single bonds and the fingerprint vibration of O-H, C-O, and C-C groups, indicative of the carbon interactions [38].

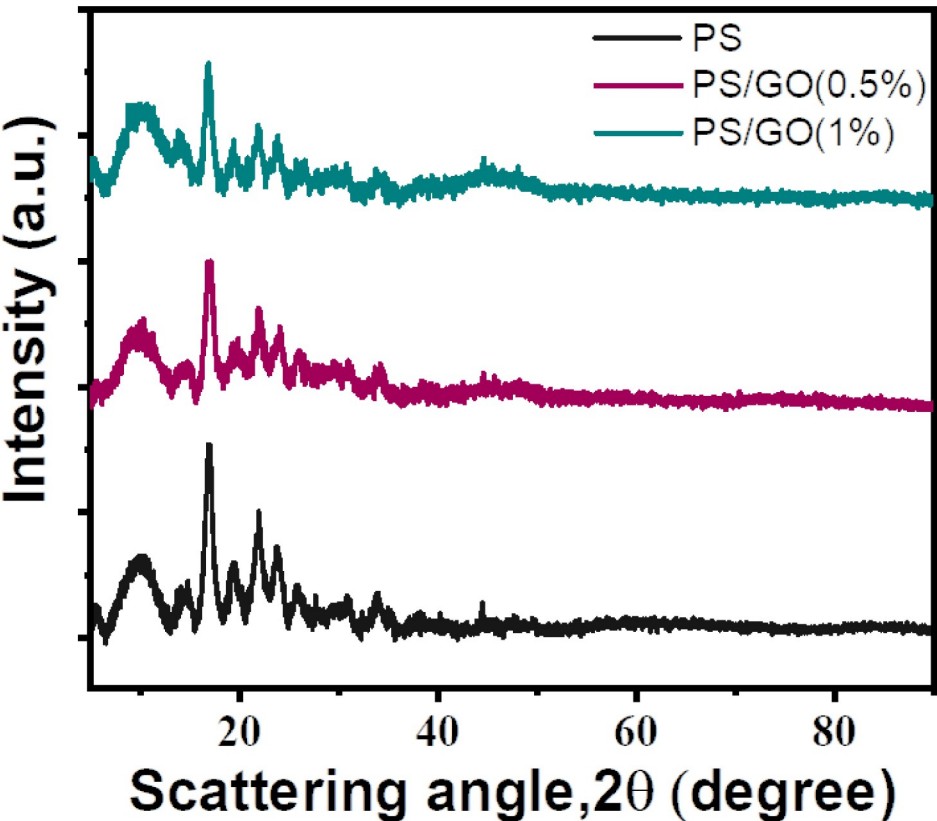

**Fig 1. XRD of PS, and PS/GO nanocomposites with concentration of GO (0%, 0.5% and 1.0%).**

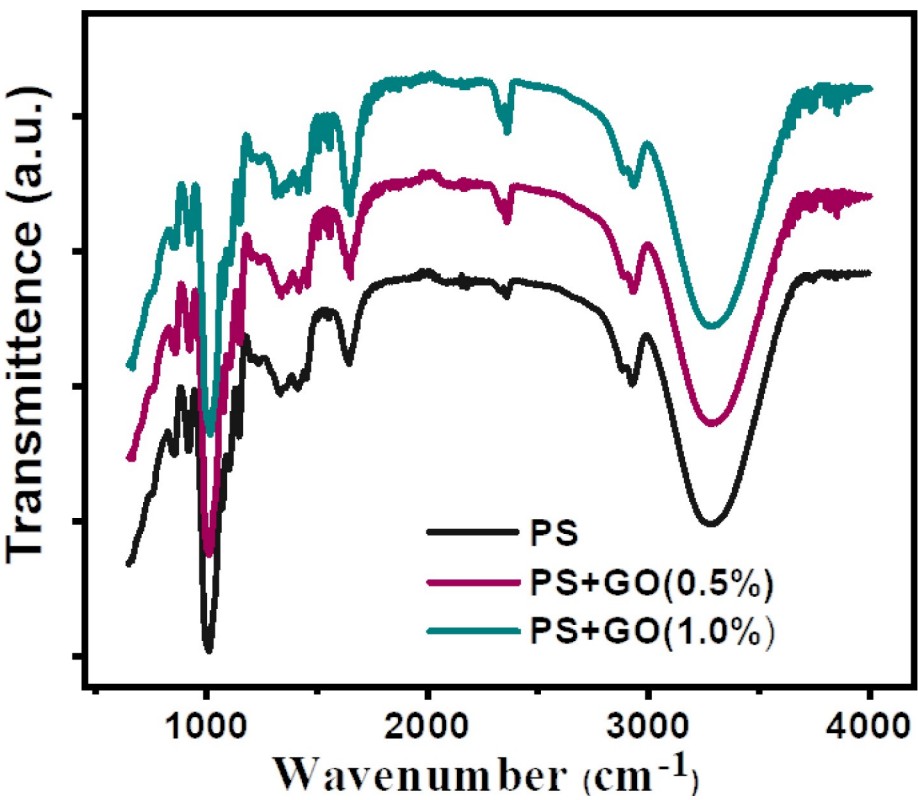

**Fig 2. FTIR spectra of PS, PS/GO nanocomposites with concentration of GO (0%, 0.5% and 1.0%).**

Apart from that, following the FTIR characteristic region, the narrow, less intense peak at 1051 cm$^{-1}$ and a peak group from 980–1300 cm$^{-1}$ may be attributed to carbon vibration with O, C, and H. The peaks observed at 1151 cm$^{-1}$ and 1109 cm$^{-1}$ denote the stretching vibration of C–O within the anhydrous glucose ring. The sharp peak at 1630 cm$^{-1}$ comes from the contribution of C = O, and C = C bond. The absence of triple bond peaks suggests the partial breaking of a long chain of amylose and amylopectin chain. The 2941 cm$^{-1}$ peak is due to the stretching of C–H bonds associated with hydrogen atoms in the ring's methane groups [39, 40].

Moreover, the intensity of the band representing the O-H group at 3352 cm$^{-1}$ rises with the inclusion of GO. The above discussion suggests the domination and recreation of carbon bonding, the replace of water molecule with GO, and also GO may take place in the affinity of water molecules [41].

### 3.3. Surface morphology analysis

The fractured surface image is obtained from FESEM. Fig 3 shows no starch residual granule for the pure PS film. With the addition and increasing concentration of GO, a distorted PS microstructure and a few lumps of agglomerations are found. Though corrugation and scrolling are inherent characteristics of graphene nanosheets, Fig 3(B) PS/GO (0.5%) suggests evenly distribution of GO throughout the PS matrix due to the effective role of branched and linear amylopectin-amylose chains and limits the shrinkage. Due to this cross-linker, the GO sheets stayed steady, and instead of getting quenched, they stacked one on another. In the image of 1% presence of GO in PS, more lumps are observed; this results from more distortion of PS, higher aggregations of GO, and inhomogeneous dispersion of carbon particles [25, 42, 43].

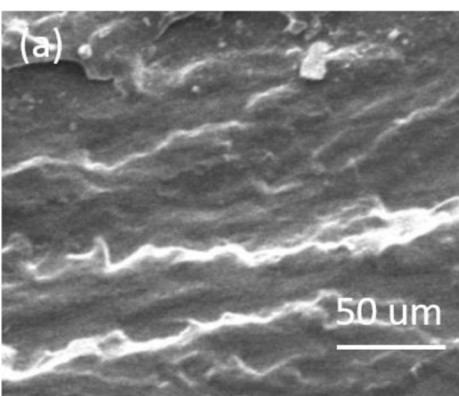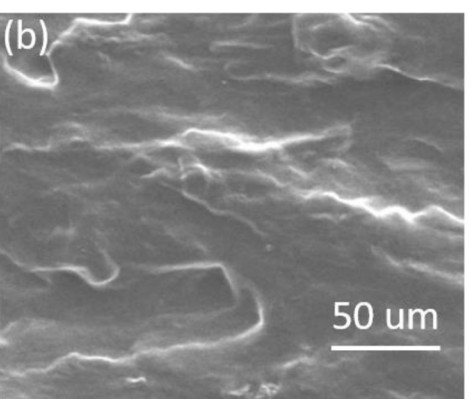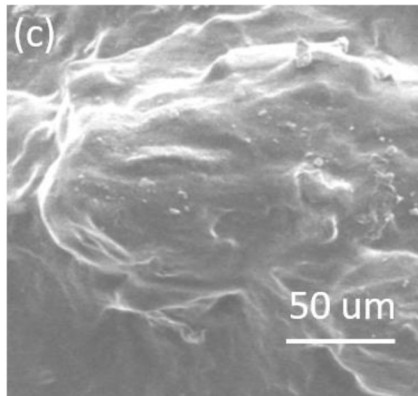

**Fig 3.** Surface morphology of (a) Pure PS (b) PS/GO (0.5%), and PS/GO (1.0%) nanocomposites.

### 3.4. Dielectric properties analysis

Dielectronic constant and loss tangent (tanδ) were calculated by the following equation, respectively:

$$C = \frac{\varepsilon_r \varepsilon_0 A}{d} \tag{1}$$

$$tan\delta = \frac{\varepsilon''}{\varepsilon'} \tag{2}$$

**3.4.1. Dielectric constant.** Fig 4 represents the effect of GO nanofiller on the dielectric constant ($\varepsilon$) of PS. For PS, the value of $\varepsilon$ does not show a large variation as the frequency changes. With the incorporation of 0.1% GO, the dielectric constant $\varepsilon$ increases nearly 1.5 times compared to pure samples, while at 0.5% loadings, it rises even higher, reaching five times that of PS.

As both $\varepsilon$ and *tanδ* are influenced by frequency, therefore electrical phenomena inside the molecular level are important to understand [44]. When the composites are subjected to an electric field, charge carriers start to move and gather at the interface. This migration occurs due to the disparity in relaxation times between the two components. The charge carriers present at the interface trigger polarization, leading to an elevated dielectric constant [45]. Defects are another reason for this increment. Defects bring changes in charge distributions in positive and negative space, which forms dipole moments, thus, enhancing the dielectric constant.

At the low-frequency region, 100 Hz, the dielectric constant increases with an additional increment of GO loading, which can be attributed to interfacial polarization occurring at the interface between the PS and GO [46]. Additionally, GO provides polar functional groups, like –COOH, –OH, –CHO, and –CO, which facilitate the interfacial and space charge polarization in the PS matrix. These functional groups also contribute to the strong bonding and ensure appropriate local electrical contacts. This, in turn, can result in interfacial polarization, playing a crucial role in the enhancement of the dielectric constant. [47]. This polarization may arise from defects/vacancies such as oxygen vacancies, surface defects, dangling bonds, and micro porosities, which act as trap states and accumulate the charge carriers. This effect is also known as the Maxwell-Wagnar effect, which works at low frequency [48, 49].

In the higher frequency range, the opposite phenomenon is observed: the dielectric constant steadily decreases as the frequency increases. The gradual decrease in the value of $\varepsilon$ with

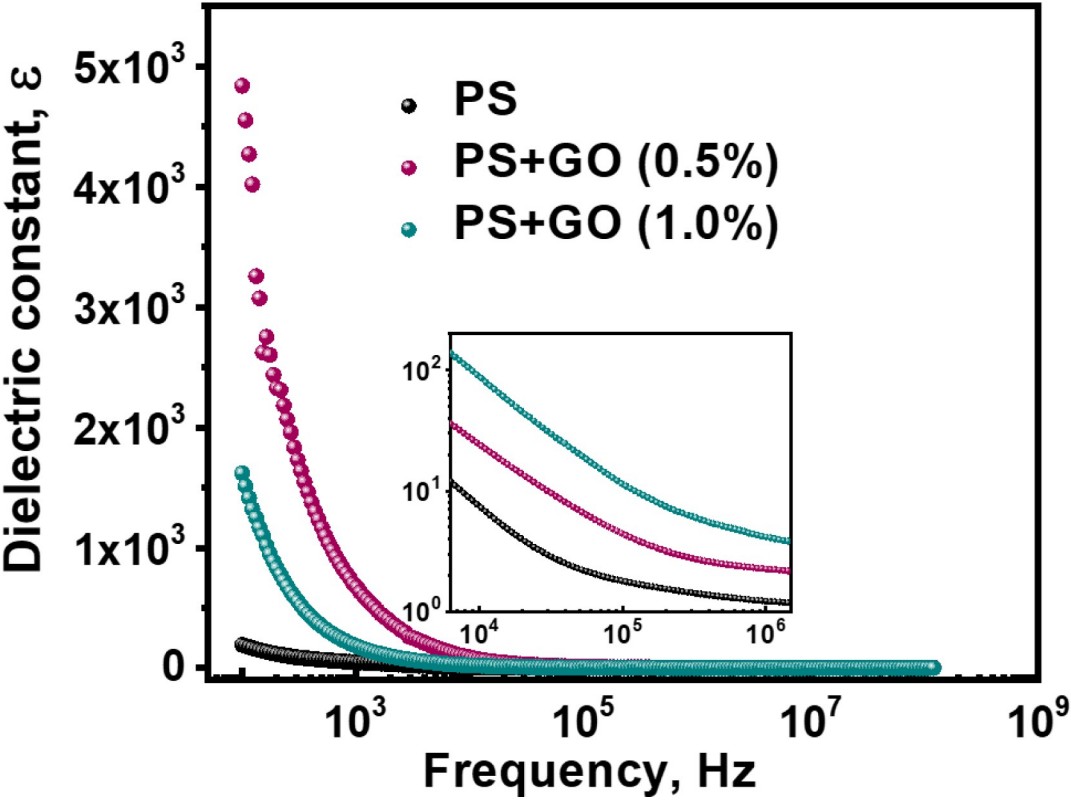

**Fig 4. Dielectric constant of PS, PS/GO nanocomposites with concentration of GO (0%, 0.5% and 1.0%) at room temperature in the frequency range 100 Hz–1 MHz.**

increasing frequency is attributed to the induced polarization process, which does not rapidly respond to changes in AC frequency. The presence of trap states leads to the generation of charges at different time constants. At higher frequencies, the induced charges are incapable of rotating rapidly enough. As a result, their frequency lags behind the applied AC signal, causing a reduction in the dielectric constant value [50, 51].

**3.4.2. Dielectric loss.** Fig 5 represents the dielectric loss of the nanocomposites as a function of the frequency. Pure PS shows a gradual loss till $10^3$ Hz, but then the loss becomes maximum in the mid-frequency, and after $10^5$ Hz, a gradual decrease is again observed. Then, for the PS/GO (0.5%) film, the dissipation factor changes in a sinusoidal way and becomes maximum at $10^2$-$10^5$ Hz but then decreases exponentially. Next, for the PS/GO (1.0%), the dissipation factor is maximum even at the lower frequency,$10^2$ to $10^5$ Hz then decrease exponentially at higher frequencies. That suggests that an increase in GO causes the incrementation of dissipation of energy as leakage current. However, at the frequency range of $10^5$ Hz, the loss is higher for 0.5wt% of GO than others [52]. This is due to the highly ordered layered structure where the trap state corrupted the order. In the high frequency, 10 MHz, the losses became minimum for all nanocomposites.

The origins of this energy loss come from Maxwell-Wagner polarization, ionic conduction, as well as dipole, electronic, and atomic mechanisms. The high losses at low frequency compared to high frequency are also a result of the interfacial polarization relaxation (Debye-type relaxation) in the composites [53]. Despite this relaxation, the leakage current near percolation is another reason for dielectric loss. Due to their low percolation threshold, the leakage current results in a significant loss factor [54].

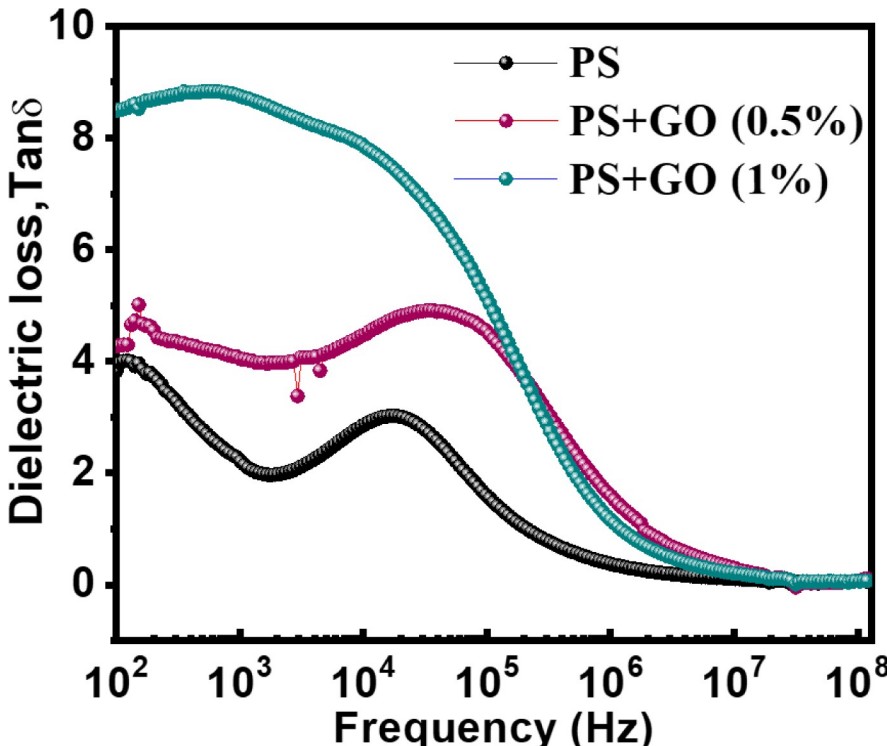

**Fig 5. Dielectric loss of PS, PS/GO nanocomposites with concentration of GO (0%, 0.5% and 1.0%) at room temperature in the frequency range 100 Hz–1 MHz.**

**3.4.3. AC conductivity.** Fig 6 represents the effects of frequency on AC electrical conductivity ($\sigma_{ac}$). All three composites showed increasing AC conductivity with increasing frequency, but comparatively lower conductivity at low frequencies. This conductivity is governed by different polarization effects. With the increase in frequency, an increase of energy is exerted on the charge carrier. At the low-frequency range, only interfacial polarization occurs between the electrode and the nanocomposite surface: the grains and other heterogeneous portions of the nanocomposite. Thus, lower conductivity is observed [50, 55]. Then, with the increase in frequency, a frequent increment of the AC conductivity is observed for all samples. The hopping or tunneling of the charge carrier takes place with an increase in frequency. The observed phenomenon can be explained by the activation of certain charge carriers that are dormant at lower frequencies but become active within this frequency range, leading to enhanced conductivity.

For 0.5% GO, a sharp increment of the conductivity is observed when compared to pure PS. The conductivity can be attributed to the increase in the formation of continuous conductive pathways within the PS/GO (0.5%) sample. However, at 1.0% filler concentrations, nanocomposites exhibit a slight conductivity improvement [53]. With the concentration increases, aggregates between GO and PS occur; thus, the bulk resistance of the composite imposes a reduction of the inter-filler tunneling may occur [56–58].

**3.4.4. Impedance spectroscopy study.** Fig 7 shows complex impedance analysis as a conventional graph of resistive data ($Z'$) against reactive data ($Z''$) with their corresponding frequency. The plotted responses appear to be different shapes in certain frequency ranges. This shape is the result of grain, grain boundary, interfacial polarization, etc. Different shapes help to separate the contribution of these active region's effects. Table 1 displays the respective

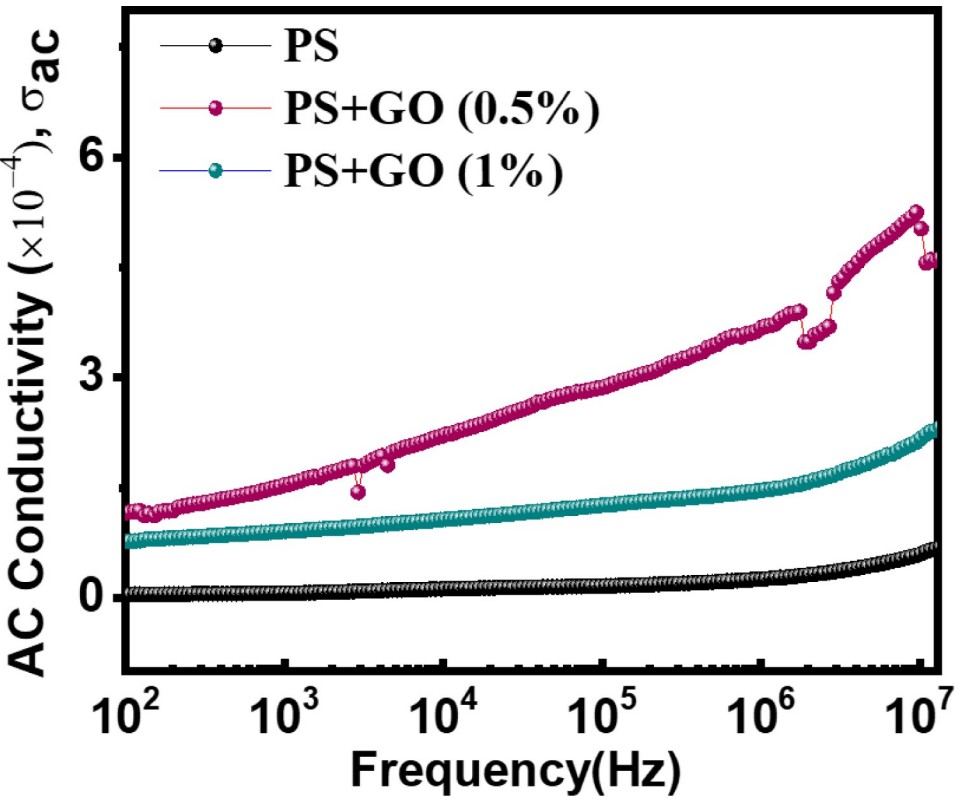

**Fig 6. AC conductivity of PS, PS/GO nanocomposites with concentration of GO (0%, 0.5% and 1.0%) within the range of 100 Hz–1 MHz frequency at room temperature.**

parameters of two distinct fitting circuits, indicating the existence of two separate relaxation mechanisms within the sample.

*3.4.4.1. Cole-Cole plot with circuit fitting.* Fig 7A illustrates the impedance spectroscopy plot, with the experimental data depicted by the dotted line and the simulation data represented by the solid line. The inset of Fig 7 displays the equivalent circuit for the nanocomposite, simulated from the plot. The simulation and circuit fitting were performed using EC lab software. The relevant parameters of the equivalent circuit are detailed in Table 2.

The equivalent circuit is consisting of all the composites and contains a series connection of two parallel circuits, with each circuit featuring a resistance (R) and a Constant Phase Element (CPE) Q, representing a capacitive component. CPE arises due to the nonideal capacitive behavior; $R_{dl}$ ($R_1$) comes from the grain contribution, and $R_{ct}$ ($R_2$) comes from the grain boundary contribution.

From Table 2, a high resistance of PS is observed, 4.3 KΩ, which comes from the rough and granule structure, which creates boundaries that interrupt the free movement of ion transportation. Additionally, PS consists of chained amylose and amylopectin, which reduce the area of ion movement. On the contrary, with the addition of GO (0.5%), the reduction of radius at a higher frequency is observed, which means the reduction of the height of the charge barrier and the decrease of the resistance to around 1.9 KΩ is also found (Table 2). The reduction of the charge barrier, facilitated by the layered structure, coupled with the presence of polar functional groups and defects acting as polarization centers, induces polarization relaxation. This process attenuates electromagnetic fields, leading to a microwave loss effect [59, 60]. Additionally, the existence of polar functional groups, hydroxyl, carboxyl, and carbonyl groups

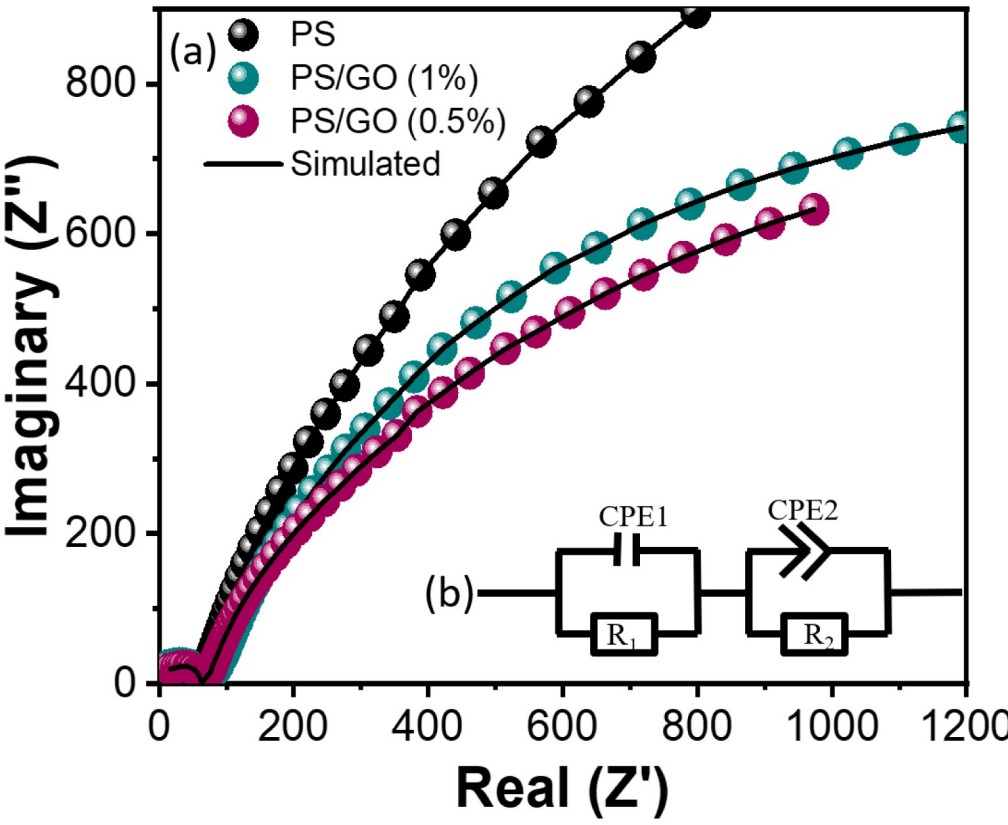

**Fig 7.** (a) Cole–Cole plots of PS, and PS/GO nanocomposites at room temperature for different amounts of GO filler incorporation (0%, 0.5% and 1.0%) (b) Fitted circuit for PS, PS/GO (0.5%) and PS/GO (1%) nanocomposites.

exhibiting varying electron affinities promotes electric dipole polarization. Consequently, the hysteresis motion of electrons within these dipoles encourages further relaxation of polarization [61–63]. Moreover, the rise in capacitance from 0.59 µF to 0.72 µF indicates the development of space charge at the electrode-electrolyte interface, referred to as the double-layer capacitive effect. The overall conductivity is primarily governed by ion conduction. The introduction of GO significantly impacts the electrical properties, and the reduction of the second semi-circle towards the left side signifies a decrease in bulk resistance, consequently leading to enhanced ionic conductivity.

**Table 1. Different types of bonding corresponding to wavenumber with the bond intensity for all the nanocomposites.**

| Bond Type | Wavenumber (cm⁻¹) | | | Bond intensity |
|---|---|---|---|---|
| | A | B | C | |
| O-H stretching | 3290 | 3290 | 3290 | Broad intense peak |
| C-H vibration | 2941 | 2941 | 2941 | Gradual decrease |
| C = O stretching | 1652 | 1652 | 1652 | Gradual increase (medium) |
| C-H bending | 1452 | 1452 | 1452 | Gradual increase (low) |
| C-O bending | 1339 | 1346 | 1326 | Medium peak |
| C-O-C asymmetric stretching | 1241 | 1241 | 1241 | Very low peak |
| O-H bending | 1018 | 1018 | 1018 | Sharp steep peak |
| C-O-C ring vibration | 865 | 865 | 865 | Low peak |

**Table 2. Parameters of the equivalent circuit components after fitting Cole–Cole curves of the PS/GO nanocomposites.**

| Composites | Electrode Resistance, R1 (MΩ) | Grain boundary Capacitance, CPE (µF) | Grain boundary resistance, $R_{ct}$ (Ω) |
|---|---|---|---|
| PS | 41.29 | 0.59 | 4300 |
| PS/GO (0.5%) | 53 | 0.72 | 1968 |
| PS/GO (1.0%) | 61 | 0.64 | 2553 |

## 4. Conclusion

Successful biodegradable PS/GO composites were synthesized. XRD indicates structural information, which shows a decrease of d-spacing in the composites with the complement of dominating carbon bonding, which was confirmed by the FTIR study. FTIR analysis also demonstrates a hydrogen bonding between PS and GO and confirms the presence of different polar functional groups. The surface morphology confirmed the layered structure of GO with PS. Then, the diminishing of grain boundary resistance and increment of AC conductivity signifies the objective of achieving a high dielectric constant together with high AC conductivity, which was achieved in the nanocomposites by the addition of GO (0.5%). Eventually, the main objective, which was ensuring a simple, low-cost synthesis process for PS/GO nanocomposites with enhanced dielectric properties, was gained. In conclusion, this synergistic PS/GO composite is promising in fabricating environmentally sustainable and biocompatible flexible electronic devices.

## Author Contributions

**Conceptualization:** Shafiqul I. Mollik, Muhammad Rakibul Islam.

**Data curation:** Eashika Mahmud, Shafiqul I. Mollik.

**Formal analysis:** Eashika Mahmud, Shafiqul I. Mollik.

**Investigation:** Eashika Mahmud, Shafiqul I. Mollik.

**Methodology:** Shafiqul I. Mollik.

**Project administration:** Muhammad Rakibul Islam.

**Resources:** Muhammad Rakibul Islam.

**Writing – original draft:** Eashika Mahmud, Shafiqul I. Mollik, Muhammad Rakibul Islam.

**Writing – review & editing:** Muhammad Rakibul Islam.

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
