## [Decision Letter · Decision Letter 0]

1 May 2024

PONE-D-24-13185Improved Dielectric Performance of Graphene Oxide reinforced Starch biopolymerPLOS ONE

Dear Dr. Islam,

Thank you for submitting your manuscript to PLOS ONE. After careful consideration, we feel that it has merit but does not fully meet PLOS ONE’s publication criteria as it currently stands. Therefore, we invite you to submit a revised version of the manuscript that addresses the points raised during the review process.

We look forward to receiving your revised manuscript.

Kind regards,

Amr Mohamed Abdelghany

Academic Editor

PLOS ONE

Reviewers' comments:

Reviewer's Responses to Questions

**Comments to the Author**

1. Is the manuscript technically sound, and do the data support the conclusions?

Reviewer #1: Yes

Reviewer #2: Partly

2. Has the statistical analysis been performed appropriately and rigorously? 

Reviewer #1: Yes

Reviewer #2: No

3. Have the authors made all data underlying the findings in their manuscript fully available?

Reviewer #1: Yes

Reviewer #2: Yes

4. Is the manuscript presented in an intelligible fashion and written in standard English?

Reviewer #1: Yes

Reviewer #2: Yes

5. Review Comments to the Author

Reviewer #1: You need to add some references in the introduction part:

1-https://doi.org/10.1016/j.carpta.2022.100239

2-https://doi.org/10.3390/polym16020294

3- DOI:10.1007/s13762-020-02764-3

4-doi: 10.3390/gels9050416

Reviewer #2: Manuscript Number: PONE-D-24-13185

Manuscript Title: Improved Dielectric Performance of Graphene Oxide reinforced Starch biopolymer

Comments:

- Title: the authors used from the first line of the abstract the expression ‘plasticized starch’ so it would be better to replace starch biopolymer to plasticized starch to be:

“Improved Dielectric Performance of Graphene Oxide Reinforced to Plasticized Starch”

And all the initials are in capital letters.

There is a great difference between starch and plasticized starch.

- Abstract: the authors started the abstract directly to the methods used and the materials. The abstract should give first a brief background about the idea/problem this work would help to solve it, the methods, results and finally the conclusion.

- Keywords: The word starch should be changed to plasticized starch.

- Novelty: The authors should emphasis the novelty of their work as there are many research articles with similar ideas and the same measurements. Moreover, they should clearly compare their works in this manuscript with that already published using GO/PS and show the advantage of GO over the reduced GO in their work and applications.

- Introduction: A little confusing and needs to be reorganized. Some abbreviations are written directly in the text without mentioning the name first. In addition, their work is focusing on the improvement of the dielectric performance of the prepared GO/PS, there is a poor information about important of this issue in the introduction

- Materials and methods: Need references in each step/paragraph. In section 2.3: ac conductivity [use capital letters; AC].

There isn’t any clear sentence that give the number of replications in each measurements or standard deviation.

- Results and discussion: Figures captions are very short. They need to be more descriptive. The discussion should contain some of the previous works that agree and also that disagree of the author’s work.

The FTIR spectra in Figure. 2 are noisy/ so thick. Moreover, a table that contains the band’s center and the corresponding band assignment should be given. This section wasn’t clear enough in the text.

Moreover,

- English language of the manuscript needs to be revised and corrected, there are many grammar mistakes, examples in the introduction section:

building a energy, a issue

- The references should be updated, some of them are very old.

6. PLOS authors have the option to publish the peer review history of their article (what does this mean?). If published, this will include your full peer review and any attached files.

Reviewer #1: No

Reviewer #2: No

---

## [Decision Letter · Decision Letter 1]

26 Jun 2024

PONE-D-24-13185R1Improved Dielectric Performance of Graphene Oxide Reinforced  Plasticized StarchPLOS ONE

Dear Dr. Islam,

Thank you for submitting your manuscript to PLOS ONE. After careful consideration, we feel that it has merit but does not fully meet PLOS ONE’s publication criteria as it currently stands. Therefore, we invite you to submit a revised version of the manuscript that addresses the points raised during the review process.

We look forward to receiving your revised manuscript.

Kind regards,

Amr Mohamed Abdelghany

Academic Editor

PLOS ONE

Journal Requirements:

Reviewers' comments:

Reviewer's Responses to Questions

**Comments to the Author**

1. If the authors have adequately addressed your comments raised in a previous round of review and you feel that this manuscript is now acceptable for publication, you may indicate that here to bypass the “Comments to the Author” section, enter your conflict of interest statement in the “Confidential to Editor” section, and submit your "Accept" recommendation.

Reviewer #2: All comments have been addressed

Reviewer #3: All comments have been addressed

2. Is the manuscript technically sound, and do the data support the conclusions?

Reviewer #2: Yes

Reviewer #3: Yes

3. Has the statistical analysis been performed appropriately and rigorously? 

Reviewer #2: N/A

Reviewer #3: N/A

4. Have the authors made all data underlying the findings in their manuscript fully available?

Reviewer #2: Yes

Reviewer #3: Yes

5. Is the manuscript presented in an intelligible fashion and written in standard English?

Reviewer #2: Yes

Reviewer #3: No

6. Review Comments to the Author

Reviewer #2: (No Response)

Reviewer #3: Dear Prof. Dr Amr

Title: Improved Dielectric Performance of Graphene Oxide Reinforced Plasticized Starch

Authors: Eashika Mahmud, Shafiqul I. Mollik, Muhammad Rakibul Islam

The authors investigate the effect of different concentrations of GO on the surface morphology, chemical, and structural properties of the PS/GO nanocomposites using FESEM, FTIR, and XRD techniques. However, several issues need improvement, including the novelty, and the clarity of language. I recommend Minor revisions.

The manuscript needs language editing: For example:

1- the 1st line in abstract , are should be is.

2- The statement “in this work……: graphene oxides to graphene oxide,, different concentration to different concentrations,, and were studied to was studied. XRD section: The diffraction peaks at……., confirms should be confirm, and this works should be this work, and so on

3- Figure 4 represents the effect of GO nanofiller on the changes of dielectric constant, ε of PS. ε for Pure PS. ε is almost similar for PS to low and high frequency.( this sentence needs to be revised, This sentence is incomprehensible

4- The manuscript needs careful linguistic review, taking into account the plural, singular, and appropriate tenses

7. PLOS authors have the option to publish the peer review history of their article (what does this mean?). If published, this will include your full peer review and any attached files.

Reviewer #2: No

Reviewer #3: No

---

## [Decision Letter · Decision Letter 2]

6 Aug 2024

Improved Dielectric Performance of Graphene Oxide Reinforced  Plasticized Starch

PONE-D-24-13185R2

Dear Dr. Islam,

We’re pleased to inform you that your manuscript has been judged scientifically suitable for publication and will be formally accepted for publication once it meets all outstanding technical requirements.

Kind regards,

Amr Mohamed Abdelghany

Academic Editor

PLOS ONE

Additional Editor Comments (optional):

I think that Authors do their best revising the manuscript and manuscript can be accepted in the current form

Reviewers' comments:

Reviewer's Responses to Questions

**Comments to the Author**

1. If the authors have adequately addressed your comments raised in a previous round of review and you feel that this manuscript is now acceptable for publication, you may indicate that here to bypass the “Comments to the Author” section, enter your conflict of interest statement in the “Confidential to Editor” section, and submit your "Accept" recommendation.

Reviewer #3: All comments have been addressed

2. Is the manuscript technically sound, and do the data support the conclusions?

Reviewer #3: Yes

3. Has the statistical analysis been performed appropriately and rigorously? 

Reviewer #3: N/A

4. Have the authors made all data underlying the findings in their manuscript fully available?

Reviewer #3: Yes

5. Is the manuscript presented in an intelligible fashion and written in standard English?

Reviewer #3: Yes

6. Review Comments to the Author

Reviewer #3: Dear Prof. Amr Mohamed Abdelghany

Regarding the manuscript PONE-D-24-13185R2, I would like to thank the authors for addressing all the comments.

The manuscript can now accepted for publication.

Best Regards

7. PLOS authors have the option to publish the peer review history of their article (what does this mean?). If published, this will include your full peer review and any attached files.

Reviewer #3: No

---

## [Editor Report · Acceptance letter]

28 Aug 2024

PONE-D-24-13185R2 

PLOS ONE

Dear Dr. Islam, 

I'm pleased to inform you that your manuscript has been deemed suitable for publication in PLOS ONE. Congratulations! Your manuscript is now being handed over to our production team.

Kind regards, 

on behalf of

Prof. Amr Mohamed Abdelghany 

Academic Editor

PLOS ONE